# Human-centred physical neuromorphics with visual brain-computer interfaces

Gao Wang ⓘ[1], Giulia Marcucci[1], Benjamin Peters ⓘ[2], Maria Chiara Braidotti ⓘ[1], Lars Muckli ⓘ[2] & Daniele Faccio ⓘ[1] ✉

Steady-state visual evoked potentials (SSVEPs) are widely used for brain-computer interfaces (BCIs) as they provide a stable and efficient means to connect the computer to the brain with a simple flickering light. Previous studies focused on low-density frequency division multiplexing techniques, i.e. typically employing one or two light-modulation frequencies during a single flickering light stimulation. Here we show that it is possible to encode information in SSVEPs excited by high-density frequency division multiplexing, involving hundreds of frequencies. We then demonstrate the ability to transmit entire images from the computer to the brain/EEG read-out in relatively short times. High-density frequency multiplexing also allows to implement a photonic neural network utilizing SSVEPs, that is applied to simple classification tasks and exhibits promising scalability properties by connecting multiple brains in series. Our findings open up new possibilities for the field of neural interfaces, holding potential for various applications, including assistive technologies and cognitive enhancements, to further improve human-machine interactions.

Visual evoked potentials (VEPs) have gained significant attention in the field of brain–computer interface (BCI) research as a promising method for establishing a direct communication pathway between the human brain and external devices[1–4]. Steady-state VEPs (SSVEPs), a class of VEPs, are neural responses generated in the visual cortex in response to periodic visual stimuli[5,6]. These responses exhibit consistent frequency components that correspond to the stimulation frequencies, making them suitable for extracting meaningful information from the brain, for example by spatially separating the lights flickering at different frequencies such that an SSVEP at one or the other frequency encodes an explicit focus of the user on a specific spatial region.

Beyond the direct encoding of light flicker frequency into an identical SSVEP frequency component, researchers have noticed that simultaneous excitation at frequencies $f_1$ and $f_2$ can lead to the generation of harmonics and additional 'inter-modulation' frequencies in general at the sum of any non-zero integer multiple of the input frequencies, e.g. at $f_1 \pm f_2$, $2f_1 \pm f_2$, etc. These frequencies are also observed to be the same for all individuals[7–11]. The full details of the physical origin of these components are still not yet fully understood but arise from the intrinsic nonlinear response of the individual neurons[12] and have recently been connected to the neuron ion channels[13].

To date, SSVEPs have been excited using low-density frequency division multiplexing (FDM) techniques, where only a few light-modulation frequencies are employed with a single flickering light stimulus[6,14–16]. The information rate of these systems is of order -1 bit/s[1], with recent work showing information transfer rates up to 5 bits/s[4]. If instead of the bit rate with which we can communicate to a computer, we consider the rate at which information can be communicated to the brain, we have a much higher -10 Mbits/s rate for the human visual system[17], implying that there may be room for applications beyond relatively simple '1-bit' BCI decision tasks. Indeed, it has been noted that whilst BCIs for machine control appear to be limited by a ceiling in terms of the information rate that actually translates into useful commands, the information transfer rate can be significantly larger and does not seem to have similar limitations[4].

[1]School of Physics & Astronomy, University of Glasgow, Glasgow G12 8QQ, UK. [2]Centre for Cognitive NeuroImaging, School of Psychology and Neuroscience, College of Medical, Veterinary and Life Sciences, University of Glasgow, 62 Hillhead Street, Glasgow G12 8QB, UK. ✉e-mail: daniele.faccio@glasgow.ac.uk

Recent work has shown that it is possible to use the human visual system for what would typically be considered as 'computational imaging' tasks, for example, a single pixel, also known as ghost imaging[18,19]. These approaches show that in principle, sufficient information for the reconstruction of grayscale images can be transmitted and processed by an SSVEP-based BCI.

In this work, we demonstrate a high-density frequency-division multiplexed SSVEP by employing hundreds of frequencies that are used to simultaneously encode and transmit information. We apply this to two examples: image transmission (used to test the efficacy of the BCI) and simple classification tasks. The latter relies on the successful implementation of a physical neural network (PNN) utilizing SSVEPs. The computation is shared between the physical visual system of the brain (for the mixing and generation of intermodulation frequencies through the intrinsic nonlinearity of the visual system) and the silicon-based computer (for the read-out and final application of weights to the data). This PNN exhibits good performance in classification tasks, demonstrating the potential of SSVEP-based BCIs for image processing and pattern recognition. Moreover, we see that classification tasks improve if we extend the SSVEP-PNN from a single-layer (i.e. a single brain) to a two-layer (i.e., two connected-brain) structure that suggests a general scalability property of our approach.

## Results

### High-density frequency division multiplexing of SSVEPs

Our experimental approach is illustrated in Fig. 1. The BCI input can be an image (e.g. a handwritten digit) that we wish to transmit and then decode via the BCI. More in general, the input data is not constrained to graphical inputs and can be, for example, a set of values that can also be combined with a set of parameters—this latter case will be used when using the BCI as a physical neural network. The information is then embedded through frequency division modulation, i.e. each pixel, $m$, in the image or, more in general, each value or parameter to be transmitted is assigned a specific light modulation frequency, $f_m$, with an amplitude $A_m$ that is given by the gray-scale value of the image pixel or by the value of the parameter to be encoded. The sum of all these frequencies,

$$x(t) = \sum_{m=0}^{M} A_m \cos(2\pi f_m t),$$ (1)

is then projected using an LED onto a white screen (a video showing the actual experiment is provided in the SM for three different cases that are described in the work). We choose the frequencies to be $f_m = f_0 + m\,\delta f$, i.e. a set of $M+1$ frequencies, each separated by $\delta f$.

We underline that in this work, we only consider the fundamental input frequency range and the intermodulation frequency range corresponding to terms of the kind $f_m + f_n$, therefore neglecting higher-order mixing terms. We also operate in what we might define as a 'narrowband' regime i.e. we always restrict the excitation input frequency band ($m\delta f$) such that the highest frequency is always significantly smaller than the second harmonic frequency $2f_0$, thus guaranteeing that we can always easily separate the intermodulation signal $2(f_0 + m\delta f)$ from the fundamental modulation SSVEP frequencies. A participant wearing an EEG device, observes the light projected onto the white screen: given the large number $M \sim 200$ of frequencies encoded into the modulated light, this will appear as a somewhat random flickering light instead of the typical periodically modulated SSVEP signal that is typically used in standard SSVEP BCIs although, it will repeat with a period $T = 1/\delta f$. The SSVEP is detected by the EEG. We use a 3-pole EEG device, which has one active electrode located in Oz (medial occipital electrode site), so as to collect SSVEP from the primary visual cortex. The EEG also has one reference electrode above the left ear, M1 position, and one ground electrode above the right ear, M2 position[19]. The detected SSVEP is a complex waveform from which we can derive a normalized power spectral distribution (NPSD), as shown in Fig. 1 and then can be used either for image reconstruction or neural network classification tasks.

### Image transmission

We investigate the information transmission capabilities of our system, specifically in transmitting black-and-white images of 14 × 14-pixel handwritten digits. We first downsample the 28 × 28-pixel handwritten digits images[20] to 14 × 14 pixels. The pixel matrix is then flattened into a linearly-indexed vector denoted as $[A_0,...,A_M]$, with length $M+1 = 196$. Since the images are black-and-white, the amplitudes $A_m$ take binary normalized values of either 0 or 1. The maximum light intensity projected by the red LED (640 nm wavelength) on the screen is 8 cd. We note that given that all frequencies are transmitted simultaneously, the final result does not depend on the rule used to assign frequencies to image pixels. The results corresponding to the transmission of the digit "7" are illustrated in Fig. 2 for $f_0 = 12$ Hz and different spacings $\delta f$ of the 196 frequencies corresponding to total SSVEP bandwidths of 1, 2, 4, 8, 12, and 16 Hz shown in Fig. 2a–f, respectively. Figure 2h shows the measured signal when the participant is blindfolded (so only the alpha

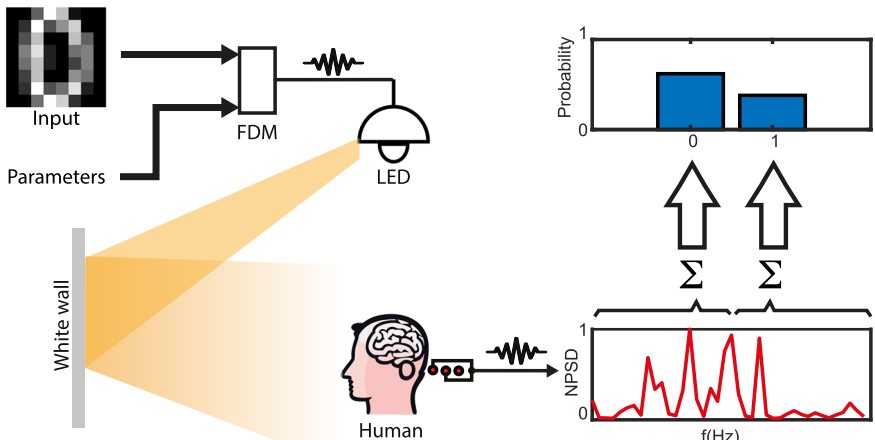

**Fig. 1 | BCI setup.** Input data (shown is an example image of a handwritten digit "0" and a set of control parameters) are encoded in frequency division multiplexing. The frequency-encoded signal modulates the intensity of an LED light projected onto a white screen, which is observed by a participant. A 3-pole EEG device detects the steady-state visual evoked potential, with an active electrode placed at Oz (medial occipital electrode site) to capture the electric signal from the primary visual cortex, a reference electrode positioned above the left ear (M1 position), and a ground electrode located above the right ear (M2 position). The resulting normalized power spectrum density (NPSD) is utilized for image transfer or computational tasks.

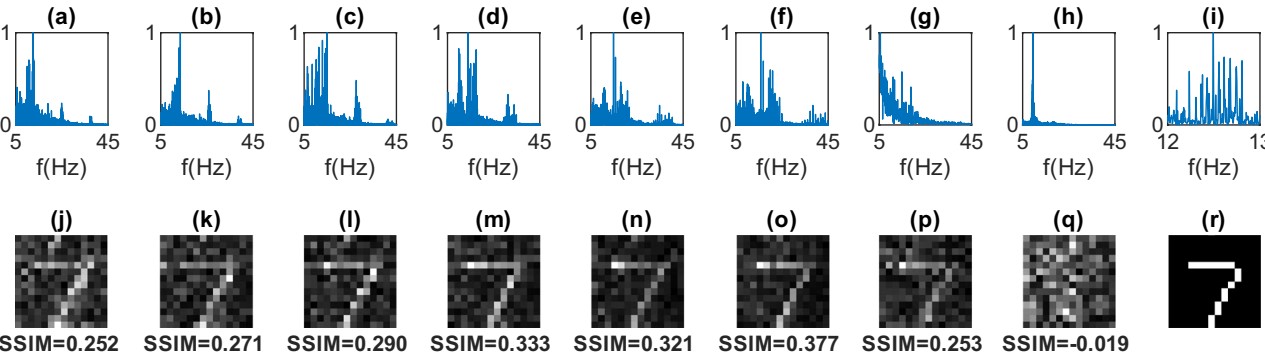

**Fig. 2 | BCI image transmission.** Experimental results are shown for a handwritten digit "7" image. The first row shows the SSVEP normalized power spectrum density (NPSD), produced by frequency division multiplexing following Eq. (1), with $f_0 = 12$ Hz, measurement time 196 s, and bandwidth **a** 1 Hz, **b** 2 Hz, **c** 4 Hz, **d** 8 Hz, **e** 12 Hz, **f** 16 Hz. **g** is for 12 Hz bandwidth and a shorter measurement time of 16.3 s, while **h** is for 12 Hz bandwidth with a blindfold (showing only an alpha wave peak at 10 Hz), and **i** is a zoom of **a** from 12 to 13 Hz. The second row, **j**–**q** shows the reconstructed, gray-scale images corresponding to the data in the image directly above in the first. Each figure also shows the structural similarity index measure (SSIM) relative to the ground truth image, shown in (**r**).

peak at 10 Hz is visible, i.e. the recorded signals in the other figures are therefore genuinely generated by the brain) and (i) shows a zoom-in of the spectrum for 1 Hz bandwidth. Figure 2j–q shows the related images that are reconstructed by simply using the amplitude of each frequency component as measured in the NPSD and reassigning it to the original pixel in the image. We note that in Fig. 2a–f, the acquisition time was 196 s, corresponding to just one full period, $T = 1/\delta f$, for the smallest bandwidth of 1 Hz. More details about the image retrieval technique can be found in the "Methods" section.

We can see that the FDM SSVEP is able to clearly reconstruct the original (ground truth) image shown in Fig. 2r, with a gradual reduction of the noise as we increase the overall bandwidth, as a result of acquiring for multiple periods (bandwidth increases from 1 to 16 Hz in Fig. 2j–o). Each image also shows the structure similarity index measure (SSIM) that quantifies the similarity of the retrieved images compared to the ground truth and indeed shows a gradual increase (improvement of the similarity) with an increasing number of measurement periods. However, if we compare the results for single period measurement times at 1 Hz (Fig. 2j) and 12 Hz (Fig. 2p) bandwidth, we see that the shorter 16.3 s acquisition time with 12 Hz bandwidth leads to a very similar reconstruction. These results indicate that there is a trade-off in terms of the signal-to-noise ratio in the transmitted data, acquisition time and bandwidth and that there seems to be an advantage in using a broad a bandwidth as possible across the EEG spectrum. These considerations need to be balanced against another effect that we see in these measurements: the neuronal response is highly nonlinear, with harmonic and intermodulation signals generated at frequencies that will overlap with the original input frequencies if the bandwidth is broader that $2 \times f_0$. In the following, we show how these intermodulation frequencies can be harnessed for more complex computations by appropriately choosing the operating bandwidths.

### Physical neural networks based on SSVEP-based BCI

To demonstrate the computational capabilities of our SSVEP-based BCI, we first show its performance in image classification. The underlying approach followed here is based on previous work on reservoir computing and extreme learning machines that can be used as a platform for machine-learning-based approaches to the classification of complex data[21–33]. Indeed, recent work has shown that by combining the physical data to be classified together with a set of control parameters or "weights" that need to be learned, one can obtain efficient classification of data in a variety of physical systems[31]. The common underlying feature of these physical systems is the requirement for a nonlinearity that will mix the input data with the control parameters

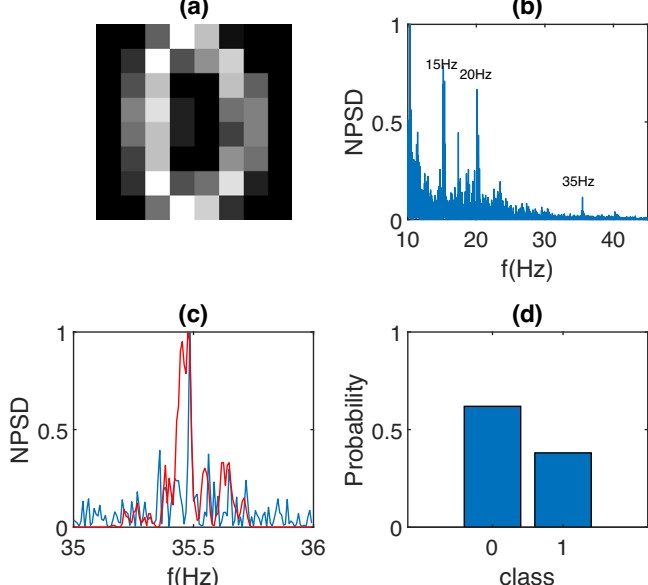

**Fig. 3 | BCI physical neural network image classification.** Experimental results from a single classification experiment of handwritten digits "0" and "1". **a** An example of input data, a grayscale $8 \times 8$ pixel digit "0". **b** Measured EEG signal NPSD with three highlighted frequency intervals: the input image frequency-encoded as 64 equidistant frequencies in the [15.0, 15.5] Hz range; the control parameters (determined by a genetic algorithm) frequency-encoded as 64 frequencies in the [20.0, 20.5] Hz range; and the 128 intermodulation frequencies in the [35, 36] Hz range. **c** The decoded intermodulation signal in more detail; the blue curve is a magnification of the measured signal in (**b**), and the red curve is the synthetic (numerically simulated) data. **d** The readout probability distribution over the two classes "0" and "1" showing a correct classification (highest probability) for "0".

combined with an algorithm that can train the control parameters so as to lead to the correct classification also of unseen data. The key point here is that the FDM-SSVEP exhibits the required features for neural network learning, i.e. we can efficiently encode relatively large amounts of information in parallel into a series of connected nodes (the different frequencies) whose mixing (i.e. the visual processing in the brain) exhibits) a strong nonlinearity.

In order to demonstrate that it is indeed possible to build a physical neural network classifier, we start with the simple task of classifying the digits "0" and "1" from a dataset of $8 \times 8$ pixel handwritten digit images[34], with an example shown in Fig. 3a. Differently from the

simpler image transmission task in Fig. 2, we now wish to perform classification of input images and, we therefore, use a physical neural network approach whereby we combine the input image information with additional control parameters that are encoded in additional frequencies. The overall frequency-encoded signal is, therefore, now defined as $X(f) + \alpha(f)$ over two distinct narrow bands, whose mathematical expression in the positive frequency domain is

$$X(f) = \sum_{m=0}^{M} A_m \delta(f - f_m), \qquad (2)$$

$$\alpha(f) = \sum_{m=M+1}^{M+P} A_m \delta(f - f_m), \qquad (3)$$

with $\delta$ the Dirac delta function. In these equations, $X$ represents the image information that is encoded into the frequency band $[f_0, f_M]$, whose frequencies $f_m$ are defined as in Eq. (1). $\alpha$ represents the control parameters that are frequency-encoded in a set of $P$ frequencies that are chosen to be larger than the image frequencies, i.e. the spectra of $x$ and $\alpha$ do not overlap. and such that the intermodulation frequency region (due to mixing between the image and control parameter frequencies) does not overlap with any linear or second-harmonic signal intervals.

Following a strategy similar to the image transmission task, the input pixel matrix is flattened into a grayscale linearly-indexed vector denoted as $[A_0, ..., A_M]$, with amplitudes $A_m$ that take normalized values in $[0, 1]$ proportional to the scale of gray tones (black corresponds to 0, white to 1). The same approach is extended to the $\alpha$-signal (full definitions are provided in the "Methods" section).

Figure 3(b) shows an example result where we chose 0.5 Hz bandwidths for both $X$ and $\alpha$ with 64 frequency components each and starting at frequencies $f_0 = 15$ and 20 Hz, respectively. Spectral components at the input frequency ranges are indicated as '15' and '20' Hz and we also highlight the intermodulation frequency band at 35 Hz that has a total bandwidth of 1 Hz. We note that the second harmonic signals at 30 and 40 Hz are only barely above the noise level, whereas the intermodulation signal appears to have a significantly better contrast.

The intermodulation signal is shown in more detail in Fig. 3c, where the blue line represents the measured NPSD, and the red curve shows the numerically simulated NPSD, with a relatively good agreement between them, indicating that the physical neural network can be optimized using synthetic data from numerical simulations. The numerical model used in this work is described in the "Methods" section, but in brief, this is based on the phenomenological observation that for a given set of input frequencies, the output SSVEP spectrum will contain the same input signals together with harmonics and intermodulation frequencies. The latter second-order frequencies are then weighted with a function $\tilde{\chi}^{(2)}$, as described in the "Methods" section, whose shape is determined from measurements and is found to be universal (i.e. independent of the EEG user) of the form $\exp(-f)$. This model can then be used to simulate a large number of different experiments using, e.g. a set of MNIST digits, which in turn are then used to learn the optimal $\alpha$ parameters with a genetic algorithm[35]. These $\alpha$ values are then used in experiments where we now collect actual EEG data from unseen examples of out-of-sample digits and perform classification.

Classification results for experimental data of unseen out-of-sample digits are shown in Fig. 3d, expressed in terms of a classification probability over the two classes "0" and "1". In general, in this work, classification probabilities are obtained by first normalizing the maximum to 1 (so as to avoid classification based on intensity) and then dividing the intermodulation frequency range into a number of segments equal to the number of classes. We then take the total power

fraction in each frequency segment to represent the classification probability.

We see that the SSVEP physical neural network is able to correctly classify the two digits. In the Supplementary Material (SM), we show more examples of handwritten digit classification together with other tasks related to different data sets, such as tumor biopsy data and classification. All these cases are relatively simple two-class classification tasks with acceptable but not exceptional classification results. We, therefore, investigated routes to improve the number of classes and classification probability.

**Multi-layer physical neural network**

One approach to improving the capability and performance of the SSVEP physical neural network could be to attempt to increase the number of nodes in the network by increasing the number of frequencies that are deployed. However, this would need to come at the expense of increasing the bandwidths and/or at the expense of increasing the frequency density. This, in turn, would lead to an increase in the required measurement time, which in the experiments described above is already 128 s. We therefore aimed for a different approach whereby we keep the single network layer as simple as possible (as shown above) and instead increase the number of layers. This is achieved by taking the outputs from the SSVEP PNN, re-encoding these by rigid translation into a new set of frequencies together with a new set of $\alpha\prime$ control parameters, and then feeding these to an LED that is observed by a second participant (schematically shown in Fig. 4). This second loop then acts as a second layer in the network—the control parameters $\alpha$ and $\alpha\prime$ in this two-layer PNN are retrained using the same numerical model used for the single layer network albeit now adjusted to account for the new multilayer structure.

Figure 4 shows results demonstrating that this 'connected brain' approach now allows us to significantly improve the classification across three classes. In this case, we chose the Iris flower dataset[36], which has five input parameters that were first resampled to 2-bit depth and then encoded into 10 1-bit (on/off) frequency components. Figure 4b shows that with just one single network layer, classification is close to chance (correct classes are indicated with gray bars), and in all cases, the overall classification probabilities are low (-50% or lower). Conversely, with a 2-layer network, classification is increased and the classification probability is also improved up to 80% and higher. We obtained very similar results across a total of 13 different individuals (3 female, 10 male, all healthy individuals, ages between 20 and 50, see SM, Fig. 5). We mentioned above that we found that the numerical model used to simulate these experiments and train the physical neural network is sufficiently simple and robust that it does not need to be tailored for different people. An immediate consequence of this is that the multilayer network approach can be implemented by either recording the output data from a single participant and then feeding the output and new parameters to the same participant or also to a different participant, therefore providing a route to connecting the brains of different people (see SM for detailed results). For the case of the PNN classification task shown in Fig. 4c), results have also been disaggregated by sex (see SM) and, with the current resolution of our tests, do not show a meaningful difference in terms of classification capability.

As a final observation, we note that although neural networks can be realized with a variety of physical systems (photonic, acoustic, hydrodynamic, etc.)[37–40], typically also with better results, the unique feature of a brain-based physical neural network is, of course, the brain and related human-element that now forms part of the system. For example, human attention can significantly modify the SSVEP response[6]. One immediate implication is that attention may, therefore, also directly modify the behavior of the PNN. We verified this explicitly by repeating the two-layer classification task spread across 6

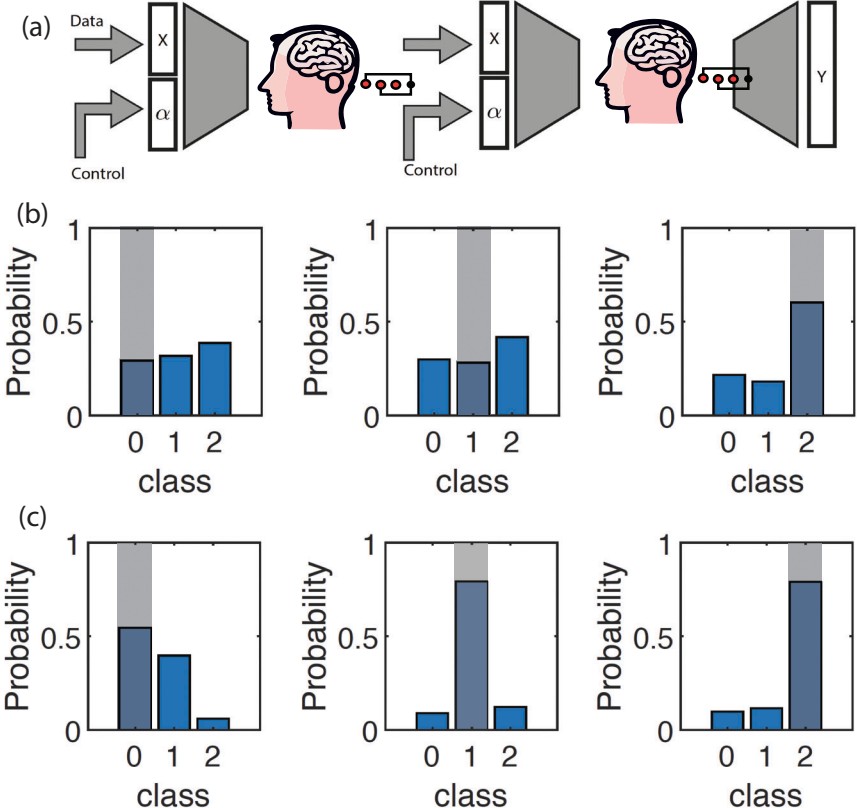

**Fig. 4 | Single and multi-layer physical neural network classification.**
**a** Schematic architecture of the two-layer PNN. **b** Classification probabilities for the single layer PNN applied to the Iris dataset with three classes. Correct classifications are indicated with gray bars. **c** Classification probabilities for the two-layer PNN applied to the same Iris dataset. All three classifications are now correct, and classification probabilities are significantly improved, from ~50% or less, now up to close to ~80%.

participants where the second participant, forming the second layer of the network, repeats measurements with an intent 'focus' on the flickering light and also, whilst still looking at the flickering light, attempts to 'disrupt' the classification by focusing their mental efforts on mathematical operations. We found a consistent reduction of the classification (Fig. 5a) and also of the amplitude of the total energy in the intermodulation frequencies (Fig. 5b). We verified this explicitly by repeating the two-layer classification task for six participants forming the second layer of the network. Participants were instructed to either 'focus' on the flickering light or, whilst still looking at the flickering light, to direct their focus internally to perform mental calculations for the 60s duration of the light stimulation, thereby 'disrupting' the focus on the flickering light. Each participant was measured twice, several minutes apart, inverting the order of the 'focus' and 'disrupt' condition, so as to exclude a possible confounding effect of the temporal order in which the conditions were performed. We found that classification accuracy ($t(5) = 6.29$, $p = 0.00006$) and the intermodulation frequency power ($t(5) = 4.18$, $p = 0.002$) were statistically significantly reduced during the 'disrupt' compared to the 'focus' condition. These results indicate that, indeed, human attention can directly modify the effectiveness of the multilayer brain connection and computing efficiency.

## Discussion

By deploying a high-density frequency multiplexing approach, it is possible to encode and transmit information in parallel over large numbers of frequency channels and tailor this encoding for specific computational tasks. These tasks are not necessarily immediately relevant for neurological studies at this stage but rather were chosen on the basis of their interest in other fields, such as physical neuromorphic computing. We have shown that image information can be transmitted with an image quality that increases for larger bandwidths.

Images are still visible for the shortest possible acquisition times, with a maximum bit-rate in Fig. 2m of (196 pixels × 1 bit/pixel)/16.3 s illumination time = 12 bits/s. A more precise evaluation based on a mutual information calculation that, therefore, rigorously accounts for pixels that are not correctly reconstructed, gives ~8.3 bits/s with a maximum bit rate across all of our measurements of 10 bits/s (see SM for full details). These values are significantly larger than the previously highest bit-rates observed of ~5 bits/s[4]. However, care is required in making these comparisons, and it is important to realize that the bit rate evaluated here is the information that can be transmitted in the form of an image encoded in a flickering light, through the brain and back to the computer, whereas typical BCI information metrics focus on the bit rate with which the brain can transmit commands to a computer. This is also the reason we preferred to resort, e.g. to mutual information estimation in order to quantify our information rates rather than use typical BCI formulas (see e.g. ref. [4]).

On the other hand, narrow bandwidth encoding over a large number of frequency channels allows to efficiently excite and isolate intermodulation frequencies that provide a route towards the implementation of SSVEP-based physical neural networks that rely on the ability to multiply the control parameters with the input data so as to create a fully connected neural layer. The underlying model for the SSVEP appears to be sufficiently simple and robust that it applies to any participant, thus implying that with the same model, we can extend this approach to connect multiple brains in a multilayered structure that improves the physical neural network capability. All of these results are also obtained in real-time, i.e. the transmitted data is analyzed in real-time, and the results are displayed whilst the participant is still attached to the EEG.

We have focused on the use of relatively narrowband input signals. Future work could instead consider the input bandwidths that are

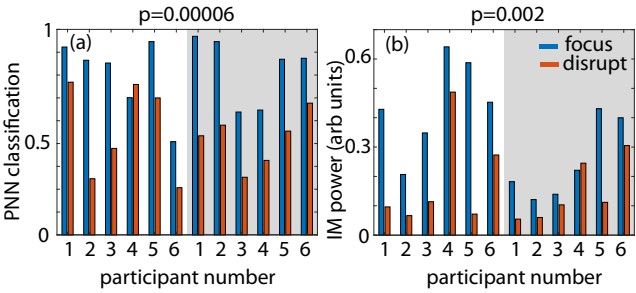

**Fig. 5 | Effect of attention on physical neural network classification and on the intermodulation (IM) frequency power. a** PNN classification probability (i.e., the power fraction in each frequency segment) and **b** intermodulation (IM) frequency power for the two layers (brain) PNN with six participants, each acting only as the second layer (the first layer is fixed, participant 1). Participants are asked to 'focus' (blue bars) attention on the light flicker or 'disrupt' (red bars) attention by mentally performing mathematical operations (number additions, subtractions, divisions) for the duration of the light flicker (200 s). In all cases, participants fixate on the illuminated area of the screen. Each participant was measured twice, several minutes apart, inverting the order of the 'focus' and 'disrupt' condition, so as to exclude a possible confounding effect of the temporal order in which the conditions were performed. We found that PNN classification accuracy ($t(5) = 6.29$, $p = 0.00006$) and the intermodulation frequency power ($t(5) = 4.18$, $p = 0.002$) were statistically significantly reduced during the 'disrupt' compared to the 'focus' condition. These results indicate that, indeed, human attention can directly modify the effectiveness of the multilayer brain connection and PNN computing efficiency.

so broad that the input frequency range overlaps with the intermodulation frequencies in the EEG signal - a more sophisticated analysis will then be required in order to de-modulate the various signals.

The advancements in SSVEP generation through high-density frequency encoding hold significant potential for applications in assistive or diagnostic technologies. By improving the robustness and scalability of SSVEP-based BCIs, we can enhance human-machine interactions, enabling individuals to interact with technology more seamlessly and efficiently. A further route for investigation will be the role of human attention in the combined human–AI computing ability also in terms of its potential for applications involving, e.g. diagnosis of attention focus and fatigue, also with extensions to the auditory paradigm. To this end, it will also be interesting to extend these studies to a broader population demographic than that tested here.

## Methods
Experiments were carried out according to ethics approval no. 300210003, University of Glasgow, with informed consent from all participants.

### EEG system
We use a three-pole EEG device, which has one active electrode located in Oz that collects the SSVEP signal from the primary visual cortex. The EEG also has one reference electrode above the left ear, M1 position, and one ground electrode above the right ear, M2 position. An audio cable leads the three-pole signal into an amplifier before being imported into the microphone port of a computer, which works as an EEG recorder.

### Light stimulus
Participants sat in front of a white screen (size: $10 \times 10$ cm) at a distance of ~50–70 cm. The white screen was illuminated by the light stimulus at an angle such that the participants could easily focus on the reflections of the light stimulus on the screen and not on the LED itself. Modulated light intensities were provided by a red LED that emits a light signal that is modulated in intensity with a modulation calculated on a computer by simply superimposing sine waves with unit or zero amplitude, according to the specific data that is encoded. The encoding schemes for each experiment are described in the main text.

Finally, before transmission from the LED, the overall total light intensity was renormalised so that the average light intensity was the same across all similar experiments and not too bright (see also supplementary videos that show examples of the light signals used in the experiment). The presented light intensities were, therefore, in the range of 0 to ~8 cd for the imaging and physical neural network classification experiments and 0 to ~1.6 cd for the second set of physical neural network experiments with focus/disrupt tests.

### Experimental procedure
Participants were unaware of the transmitted stimuli and were only instructed to fixate on the center of the white screen throughout the stimulation period. During the attention experiment, participants were, before each block either instructed to focus on the light ('focus' condition) or to perform mental calculations ('disrupt' condition). Each block of stimulation had a duration equal to the inverse of the frequency separation of the illumination signals, as described in the main text. Between blocks, participants could rest before starting the next block.

### SSVEP image retrieval
As mentioned in the main manuscript, the EEG measurements corresponding to Fig. 2a–e have a measurement time of $T = 196$ s, which equals the reciprocal of the frequency resolution, i.e., $T = \frac{1}{\delta f}$ for the encoding signal with the longest period (Fig. 2a) with 1 Hz bandwidth. In contrast, the measurement for Fig. 2f is conducted with a measurement time of $T = 16.3$ s (12 Hz bandwidth). Furthermore, the reported EEG output spectra in Fig. 2a–g are all normalized with respect to the maximum output signal value, a convention adhered to throughout this paper.

The second row of Fig. 2 displays the reconstructed images in grayscale. Here, white represents the NPSD normalization constant, black denotes the zero value, and intermediate gray tones are determined proportionally. In other words, if $x$ is the PSD at one of the encoding frequencies, the corresponding gray tone is calculated as $\frac{x}{x_{max}}$.

This specific task, due to its lower complexity, stands as the sole instance within this work where both encoding and decoding processes employ an identical vector quantization strategy. This entails representing them as linearly-indexed vectors of amplitudes associated with matching frequency distributions. Indeed, from the SSVEP NPSDs in Fig. 2a–f, one can observe the emergence of higher-order harmonics, with a notable emphasis on sum frequency generation (SFG) resonances. Subsequently, to facilitate a high level of information integration in more complex tasks, we choose to carry out the PNN decoding in the SFG regime.

### SSVEP training model
We build a phenomenological model of our SSVEP experiment, inspired by standard models in nonlinear optics[41].

In the frequency domain, Eq. (1) becomes

$$\tilde{x}(\omega) = \pi \sum_{m=0}^{M} A_m \left[ \delta(\omega - \omega_m) + \delta(\omega + \omega_m) \right], \tag{4}$$

with the Fourier transform (FT)

$$\tilde{x}(\omega) = \int_{-\infty}^{+\infty} dt\, x(t) e^{i\omega t}, \tag{5}$$

the inverse Fourier transform (IFT)

$$x(t) = \frac{1}{2\pi} \int_{-\infty}^{+\infty} d\omega\, \tilde{x}(\omega) e^{-i\omega t}, \tag{6}$$

and $\omega = 2\pi f$.

We define the EEG signal as

$$\tilde{y}(\omega) = \sum_{n=1}^{N} \tilde{\chi}^{(n)}(\omega)\mathcal{F}[x^n](\omega) = \sum_{n=1}^{N} \tilde{y}^{(n)}(\omega), \tag{7}$$

with $\tilde{\chi}^{(n)}(\omega)$ the $n$th order harmonic amplitude, and $\mathcal{F}$ standing for the FT operator. In some realizations of our BCI, we add a quadratic activation function in the readout, getting a final output

$$y_{\text{out}}(\omega) = ||y(\omega)||^2. \tag{8}$$

Starting from Eq. (1) with $M \geq 1$ and FDM defined as

$$0 \ll \omega_a - \frac{d\omega}{2} \leq \omega_0 \leq \omega_1 \leq \ldots \leq \omega_M \leq \omega_a + \frac{d\omega}{2}, \tag{9}$$

such that it is confined into a narrow band, we get an SFG profile as

$$Y_{SFG}(\omega) = \frac{\pi}{2}\tilde{\chi}^{(2)}(\omega)[X * X](\omega). \tag{10}$$

More details about the derivation of the latter equation are reported in the Supplementary Material.

To consider two distinct narrow bands, we define

$$0 \ll \omega_a - \frac{d\omega}{2} \leq \omega_0 \leq \ldots \leq \omega_M \leq \omega_a + \frac{d\omega}{2}, \tag{11}$$

$$\omega_a + d\omega < \omega_b < 2\omega_a - d\omega, \tag{12}$$

$$\omega_b - \frac{d\omega}{2} \leq \omega_{M+1} \leq \ldots \leq \omega_{M+P} \leq \omega_b + \frac{d\omega}{2}. \tag{13}$$

The corresponding input spectrum, once the difference frequency generation (DFG) terms are removed and we impose $\omega > d\omega$, is defined by $X(\omega) + \alpha(\omega) = \sum_{m=0}^{M+P} A_m \delta(\omega - \omega_m)$, with $X(\omega)$ and $\alpha(\omega)$ defined as

$$X(\omega) = \sum_{m=0}^{M} A_m \delta(\omega - \omega_m), \tag{14}$$

$$\alpha(\omega) = \sum_{m=M+1}^{M+P} A_m \delta(\omega - \omega_m), \tag{15}$$

as in Eq (3), with the added constraint that $\sum_{m=0}^{M} A_m^2 = \sum_{m=M+1}^{M+P} A_m^2 = 1$ to ensure a balanced distribution of amplitudes between the two signals.

Equations (10) and (15) imply that the pure SFG profile, generated by sums of one input and one control frequency, is now given by

$$Y_{X+\alpha}(\omega) = \frac{\pi}{2}\tilde{\chi}^{(2)}(\omega)[X * \alpha](\omega), \tag{16}$$

as proven by Eq. (6) in the Supplementary Material.

We emphasize that despite the apparent linearity of the equation, its physical nature remains inherently nonlinear, primarily due to overall amplitude coupling, as all amplitudes originate from the same LED source. To illustrate the implications of this nonlinearity in mathematical terms, consider a scenario where these signals are influenced by perturbative white noise denoted as $\xi$, and/or the LED intensity is amplified by a factor $\sigma$. In this situation, both of our signals transform as follows:

$$X' = \sigma X + \xi, \quad \alpha' = \sigma\alpha + \xi. \tag{17}$$

Subsequently, Eq. (16) can be expressed as

$$Y_{X'+\alpha'} = \frac{\pi}{2}\tilde{\chi}^{(2)}[\sigma^2 X * \alpha + \sigma(X+\alpha) * \xi], \tag{18}$$

This formulation highlights the enhanced mixing and randomness capabilities inherent in our processing stage compared to a simpler linear reservoir.

**Reporting summary**

Further information on research design is available in the Nature Portfolio Reporting Summary linked to this article.

## Data availability

Data and code relevant to this work are deposited at https://doi.org/10.5525/gla.researchdata.1640. The data are available under restricted access for ethics reasons, access can be obtained by contacting the authors and available upon request.

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

## Acknowledgements

The authors acknowledge discussions with Angus Paton and financial support from the UK Engineering and Physical Sciences Research Council (grant EP/T021020/1). G.W. acknowledges the support of the China Scholarship Council. D.F. acknowledges support from the Royal Academy of Engineering Chair in the Emerging Technologies program. B.P. has received funding from the EU Horizon 2020 research and innovation program under the Marie Skłodowska-Curie grant agreement no. 841578.

## Author contributions

G. Wang: conceptualization, experiments, data acquisition, data analysis, original draft preparation. G. Marcucci: data analysis, supervision, original draft preparation. B. Peters: conceptualization, data analysis, writing—reviewing and editing. M.C. Braidotti: supervision, writing—reviewing and editing. L. Muckli: conceptualization, writing—reviewing and editing. D. Faccio: conceptualization, supervision, original draft preparation.

## Competing interests

The authors declare no competing interests.
