## [Peer Review File · Nature Communications]

Human-Centred Physical Neuromorphics with Visual Brain-Computer Interfaces.Reviewers' comments:

Reviewer #1 (Remarks to the Author):

In this study, the authors show that information can be encoded in high-density frequency-division multiplexing excited SSVEPs. Although research in this area should be encouraged, improvements are needed in the current version of this study.

This study only used 6 subjects to verify the feasibility of the proposed system. Compared with existing studies, the number of subjects in this study is relatively small, and it is recommended to increase the number of subjects.

Hundreds of stimulus frequencies are presented at the same time, how to determine the corresponding intermodulation frequency components? Whether the intermodulation frequency components remain consistent for different subjects? Please discuss.

For different handwritten digit images, whether the stimulation frequencies and the distribution of stimulation frequencies are consistent? Please specify.

In this study, the authors use hundreds of stimulation frequencies to encode images. Is there any room for optimization of the number of stimulation frequencies, or to study the relationship between the number of stimulation frequencies and recognition performance?

Online system testing is the gold standard for evaluating the performance of brain-computer interfaces. This article is all offline analysis, and no online experiments are designed to verify the system performance. The feasibility of the system still needs to be further verified.

Frequency encoding and intermodulation frequency components have been reported in the existing SSVEP-BCI literature, and the author did not clearly explain the significance of using hundreds of stimulation frequencies to encode, so the innovation of this article is limited.

Reviewer #2 (Remarks to the Author):

In this manuscript, the authors described an experiment where they leveraged the visual systems of human subjects for information processing, such as performing machine learning tasks. In a nutshell, the experimental scheme is as follows: the information to be processed is encoded as time-varying light intensity of different frequencies (each frequency bin encodes a number, and they are slow enough for human visual systems to be responsive.). This flickering light was displayed to test human subjects, and was used to induce brain activity through a mechanism called steady-state visual evoked potential. The evoked brain activities were measured through single-point EEG as electric signals, and then the EEG signal was read out and further processed to get the result of the task (e.g., to decode the image shown to the subject or to classify different classes of images shown to the human subject.) The concept of the work is intriguing, and I wish to ask questions about experimental details in order to better understand the work:

1. There are two essential questions: where did the signal processing take place? How sophisticated was the processing? While the manuscript has shown a series of interesting experiments, they still did not clearly provide answers to the two questions above. The following is my attempt to break down the questions.

2. Did the authors make sure there is no information leakage in the experiment, in the sense that there is no information pathway that directly passes the information from the LED display to the EEG while skipping the human subject altogether? For example, if the human head were replaced with a dummy head, would the result be the same? Many mechanisms could "short-circuit" information from LED to EEG. For instance, if they share the same power supply, the switching of the LED can generate a strong enough signal for the EEG to pick up if there is poor electrical insulation. I am asking the question just to make sure we can ensure at least part of the computation took place in the head of the human subjects. If we can verify this point, I would say the experiment is at least non-trivial since there is at least some signal transduction from the optical inputs to the human visual system to electrical activities in the human brain.

3. The authors have used the generation of new frequencies through nonlinear wave mixing to illustrate the computational sophistication (a practical use of this wave mixing is it allows us to inject additional parameters to be mixed with the input data, to better control the machine learning processing, instead of treating the physical system as a completely random reservoir.) Meanwhile, the authors seem to suggest the same wave mixing model applies to all the six different human participants involved in the study, if I understand it correctly. This is somewhat surprising, since if the mechanism of nonlinearity indeed results from the operation of biological neurons, as the authors suggested, I would assume some degree of person-to-person variation. I am curious about the level of variation observed in this experiment. Is it possible that the nonlinear mixing actually did not happen in the human brain, but instead is due to some nonlinear effects of the recording equipment of EEG used for all experiments?

4. About latent variables: related to the question of person-to-person variation, human subjects tend to have quite strong latent states, including mood and mental states, and therefore it is typically challenging to control all the experimental variables and strictly repeat the experiment, sometimes even on the same subject. As the authors have correctly identified attention as a factor that affects experiment outcome, I wish to learn about other possibilities. For example, could any subject identify which hand-written digit was shown just by looking at the flickering light? If the subject already had an answer in his/her mind, the EEG might just be reading out that answer in some binary choice manner. This would be acceptable, since it means cognition also played a role in this task, in addition to some simple mechanics of human visual systems.

5. A technical question: what is the highest modulation frequency the human visual system can respond to? Is there a frequency transfer function?

6. A technical question: The image encoding scheme seems to hide the correlation between adjacent pixels in a 2D image, and therefore worth more consideration. A natural question is: what if we permute all the pixels and encode them with different frequencies? The resultant light flickering pattern would be different, but it still represents the same image. I am curious to know how the classification results would be.

7. A technical question: in some binary classification problems, such as differentiating 0 and 1, despite the flickering pattern, I think the total intensity of the light is definitely quite different if both images are normalized only to the brightest pixel in each image. This is because images of 0s tend to have many more bright pixels than images of 1s. I suspect humans can classify these two digits just based on average intensity, and therefore I would be curious to know what the result would be if images are normalized to have the same average intensity.

8. Another technical question: I still find it hard to reconcile the fact that the level of attention affects classification performance while the nonlinear wave mixing model does not seem to vary as much. I wish to verify if attention changes the nonlinear wave mixing model (i.e., χ_2 , also it is also curious to understand why a single χ_2 works for all pairs of frequencies.).

9. There are several typos in the manuscript. For example, there are two (m) panels in Figure 2, while one of them should be (g). There is 'network' spelled as 'netwoek', and so on. The authors might want to carefully check the manuscript and reference list again.

Overall, the work is novel, and it would be a good contribution to the physical computing community if we can draw more definite conclusions on what we learned from these experiments.

Reviewer #3 (Remarks to the Author):

This paper proposed a method to encode and decode image data through BCI. While this paper presents interesting work and idea, the practicality of such method remains questionable.

The stimulation/recording time is too long (16 to 196 s) compared to traditional SSVEP-based BCI (typically <3-5 s). It will be even longer if the image is coloured and/or more complex (more pixels, higher resolution).

The proposed method could be very useful in image transmission with no prior knowledge on the

image. However, the current structure of the proposed approach still relies on image identification, which requires knowledge to the potential image set. In this case, labelling each possibility with a frequency and perform a traditional SSVEP task might be a more practical solution.

This proposed method also poses a high requirement on the stimulation system, and the authors have not discussed this aspect in detail.

Comments:

1. The paper stressed on the low ITR in current solutions. But the proposed method did not solve this problem. The ITR from this work was not reported.
2. The authors should keep their notations consistent (e.g., on page 2 right below equation 1, the use of 'm' and 'M') and explain all parameters.
3. When mentioning Fig.2f, the single period stimulation time should be longer than 16s. It is inaccurate and misleading to say $T=16s$.
4. The 196 s stimulation is really long. It is very hard for the users to stay focused on the visual stimulation for over 3 minutes. The stimulation may also cause severe visual fatigue.
5. Page 3 last sentence in the first paragraph: remove the extra ')'
6. Missing details on EEG system used in the experiment.
7. Missing details on experimental setup, especially the stimulation system.

Reviewer #1 (Remarks to the Author):

In this study, the authors show that information can be encoded in high-density frequency-division multiplexing excited SSVEPs. Although research in this area should be encouraged, improvements are needed in the current version of this study.

This study only used 6 subjects to verify the feasibility of the proposed system. Compared with existing studies, the number of subjects in this study is relatively small, and it is recommended to increase the number of subjects.

REPLY: The presence of a strong underlying effect in the population can, in principle, be shown statistically with very few participants. However, we agree that given the novelty of these findings, we need to provide confidence that they were not produced by chance. We have now therefore replicated our findings in an independent set of seven new subjects. The total subjects tested is now 13. We have added a new figure in the supplementary material that shows for example a box plot summarising the results from all 13 participants on the PNN classification task. As can be seen, the results are consistent for all 13 participants. We also note that we have not excluded any participants from the results, i.e. our method has worked for all tested people. This replication supports that this effect exists in the population and is not dependent on the sampled participants. We see no change in the results, which we think is very promising. We are therefore confident that our results hold for the whole population we sampled in our experiments. However, explicitly comment in the conclusion, that future studies need to validate the effect in subpopulations (e.g., patients or elderly) in which the approach might be implemented. We do however underline that the scope here is not to demonstrate a new medical or neuroscience application but simply to show that it is possible to densely multiplex signals in SSVEP and use this for other tasks beyond standard BCI applications.

Hundreds of stimulus frequencies are presented at the same time, how to determine the corresponding intermodulation frequency components? Whether the intermodulation frequency components remain consistent for different subjects? Please discuss.

REPLY: We follow very standard procedure for BCI using SSVEP, i.e. we take a Fourier transform of the signal which gives us all of the spectral components. We know the frequencies that are encoded in the original signal, e.g. ω_1 , ω_2 etc. And we see these in the spectrum. But we then also see signals at combinations of these frequencies, e.g. $\omega_1 + \omega_2$ etc. These are defined in the literature as “intermodulation frequencies”. This relation is mathematical and cannot deviate from these predicted values. So indeed, they are absolutely consistent across all subjects. This is a relatively well reported effect in the literature that we also cite in our manuscript. We have added an explicit comment that these frequencies are fixed for all individuals in the introduction where we introduce the concept of the intermodulation frequencies.

For different handwritten digit images, whether the stimulation frequencies and the distribution of stimulation frequencies are consistent? Please specify.

REPLY: yes, we have used the same encoding scheme for all the imaging experiments and details e.g. for the frequency bands and frequency separations are provided in the paper. For example, in Figure 2 we explicitly show the effect of using different frequency bands and separations.

In this study, the authors use hundreds of stimulation frequencies to encode images. Is there any room for optimization of the number of stimulation frequencies, or to study the relationship between the number of stimulation frequencies and recognition performance?

REPLY: we briefly discuss and have expanded on this in the revisions. We need to make sure that the input frequencies do not overlap with any potential intermodulation frequencies as it then becomes very difficult to distinguish if a given frequency signal amplitude is due to the direct stimulation at the frequency or if it is generated also by an intermodulation effect. We now explicitly mention that we always work in a “narrowband” regime where the input frequency bandwidth is small (specifically, smaller than $2*f_0$), i.e. such that the input frequencies can never overlap with the intermodulation frequencies. We also mention in the conclusions that we will investigate in detail what happens in this case in future work (ongoing) as this case is outside the scope of the current study.

Online system testing is the gold standard for evaluating the performance of brain-computer interfaces. This article is all offline analysis, and no online experiments are designed to verify the system performance. The feasibility of the system still needs to be further verified.

REPLY: our research is not primarily focused on BCI development for the real-time control of machines or devices. Rather, we intend BCI in the more generic sense where a computer and a human can interact. What we are hoping to show here is that there is a new way to do this by using multiple frequencies. That said, we should underline that experiments are indeed carried out in real time. For example, the training of the neural network is performed offline before the experiments and only used synthetic data (i.e. it does not require experimental input). The inference can be very fast and is indeed performed whilst the subject is still attached to the EEG. We have now included a sentence in the conclusions explaining this.

Frequency encoding and intermodulation frequency components have been reported in the existing SSVEP-BCI literature, and the author did not clearly explain the significance of using hundreds of stimulation frequencies to encode, so the innovation of this article is limited.

REPLY: we have cited this literature and we have explained how most SSVEP systems use 2 or 3 frequencies. We provide an analogy with optical telecommunications that moved from the original single frequency for information transport to multiple frequency, thus enabling the parallel transmission along many many channels in parallel. So our approach is basically a multi-channel SSVEP system that has not been considered before. We then show applications such as image transmission where each frequency encodes one pixel and we can therefore transmit complex information such as images (so standard SSVEP would only provide images 2-3 pixels in parallel) or even information required for machine learning applications. We have modified the text so that we now also explicitly note in the conclusions that this approach has also allowed us to improve the bit-rate of standard BCI systems by an order of

magnitude (1 bit to 12 bits) and is more than 2x times larger than the previously highest bit-rate reported.

Summarising: we thank the referee for their comments. We hope that we have addressed their concerns and that we have now provided the information required for a more general audience to appreciate our work and results. We do think that the manuscript has benefitted from this and we hope that the referee can agree with us.

Reviewer #2 (Remarks to the Author):

In this manuscript, the authors described an experiment where they leveraged the visual systems of human subjects for information processing, such as performing machine learning tasks. In a nutshell, the experimental scheme is as follows: the information to be processed is encoded as time-varying light intensity of different frequencies (each frequency bin encodes a number, and they are slow enough for human visual systems to be responsive.). This flickering light was displayed to test human subjects, and was used to induce brain activity through a mechanism called steady-state visual evoked potential. The evoked brain activities were measured through single-point EEG as electric signals, and then the EEG signal was read out and further processed to get the result of the task (e.g., to decode the image shown to the subject or to classify different classes of images shown to the human subject). The concept of the work is intriguing, and I wish to ask questions about experimental details in order to better understand the work

1. There are two essential questions: where did the signal processing take place? How sophisticated was the processing? While the manuscript has shown a series of interesting experiments, they still did not clearly provide answers to the two questions above. The following is my attempt to break down the questions.

REPLY: We believe that the referee has understood correctly – part of the computation is performed by the visual system and part is performed by the computer. For the brain part, we adapt standard SSVEP techniques in this work: a modulated light source is observed by the subject. This induces a neural response in the visual cortex that is picked up an EEG electrode. The novelty here is that instead of one or two frequencies, we now use hundreds in parallel and computing occurs in the form of frequency signal mixing followed by the measurement and use of these intermodulation EEG frequencies. The recorded EEG signal is then processed on a computer. This computer-based processing is then simply a Fourier Transform of the signal that will give the SSVEP spectrum. We then, by combining these two steps, obtain images or perform machine learning. We have added a sentence that explains this in the revised text (end of introduction section).

2. Did the authors make sure there is no information leakage in the experiment, in the sense that there is no information pathway that directly passes the information from the LED display to the EEG while skipping the human subject altogether? For example, if the human head were replaced with a dummy head, would the result be the same? Many mechanisms could "short-circuit" information from LED to EEG. For instance, if they share the same power supply, the switching of the LED can generate a strong enough signal for the EEG to pick up if

there is poor electrical insulation. I am asking the question just to make sure we can ensure at least part of the computation took place in the head of the human subjects. If we can verify this point, I would say the experiment is at least non-trivial since there is at least some signal transduction from the optical inputs to the human visual system to electrical activities in the human brain.

REPLY: We understand the referee's concern and can reassure our readers that indeed the intermodulation frequency mixing is occurring in the brain and is not an artefact. We have explicitly verified this by ensuring that there is no possibility for short-circuiting the various parts of the experiments. The intermodulation frequencies are most definitely generated inside the head and removing the head simply gives no output from the EEG device. We thought that it might be instructive to include a new figure in the paper, Figure 2(g) and (o) where we explicitly show this by repeating the imaging experiment with a blindfolded person. This has the benefit of: (a) showing that all of the SSVEP signals completely disappear, and (b) shows that the EEG is still working correctly by showing that we still observe the expected alpha waves at 10 Hz. Figure (g) shows the actual measured signal. Figure (o) shows the attempt to reconstruct an image which of course fails as we only have noise at the SSVEP frequencies as these can no longer be processed by the visual system when blindfolded.

3. The authors have used the generation of new frequencies through nonlinear wave mixing to illustrate the computational sophistication (a practical use of this wave mixing is it allows us to inject additional parameters to be mixed with the input data, to better control the machine learning processing, instead of treating the physical system as a completely random reservoir.) Meanwhile, the authors seem to suggest the same wave mixing model applies to all the six different human participants involved in the study, if I understand it correctly. This is somewhat surprising, since if the mechanism of nonlinearity indeed results from the operation of biological neurons, as the authors suggested, I would assume some degree of person-to-person variation. I am curious about the level of variation observed in this experiment. Is it possible that the nonlinear mixing actually did not happen in the human brain, but instead is due to some nonlinear effects of the recording equipment of EEG used for all experiments?

REPLY: This is an interesting question. One needs to bear in mind a few points though. First of all, the intermodulation frequencies can only appear at exactly the frequencies that are sums of the input frequencies, i.e. $f_m + f_n$. This is a very generic result of any nonlinear system. Said differently, an input sinusoidal signal is recorded by the retina and transmitted to the visual cortex. In this process, the signal can become distorted. Physically, the Hodgkin-Huxley model for a single neuron shows nonlinear behaviour, so it is maybe not surprising that the visual system is also nonlinear. There is also recent work that studies in detail the origin of the nonlinearity and this indeed resides in the physics of neuronal response and is therefore a very general effect that should appear in the same way in all people [Luff et al., *bioRxiv* (2024) <https://doi.org/10.1101/2023.01.05.522833>]. Standard Fourier theory tells us that the distorted signal will be a sum of the original input frequency plus weighted harmonics, i.e. second, third etc harmonics. This amounts to saying that the distortion is related to a nonlinearity and the same nonlinearity can be responsible for mixing/summation of two frequencies f_m and f_n .

The distortion of the input signal is therefore a generic feature of the visual system pathways and does not originate e.g. in the specific neural connectivity of individual neurons. For this reason, the frequency mixing process is also generic and is observed to be the same in all people – it is a generic feature of neuronal transmission in vision. That said there is some variability from one person to another – this is clear in Figure 5 for example where the classification amplitude depends on the individual and this is a result of the amplitudes of the intermodulation signals varying slightly from one individual to the next. However, the existence of these frequencies and the generic behaviour, does not depend on the individual. We have added more references in the introduction section together a more extended description of the nonlinearity. We hope this clarifies the point that this nonlinear response is known and expected and it is not due to the recording equipment (that we have also ruled out by the additional experiments that have been added in reply to comments above).

4. About latent variables: related to the question of person-to-person variation, human subjects tend to have quite strong latent states, including mood and mental states, and therefore it is typically challenging to control all the experimental variables and strictly repeat the experiment, sometimes even on the same subject. As the authors have correctly identified attention as a factor that affects experiment outcome, I wish to learn about other possibilities. For example, could any subject identify which hand-written digit was shown just by looking at the flickering light? If the subject already had an answer in his/her mind, the EEG might just be reading out that answer in some binary choice manner. This would be acceptable, since it means cognition also played a role in this task, in addition to some simple mechanics of human visual systems.

REPLY: all tests were blind, i.e. the subject does not know what information is being transmitted. But beyond this, the real point here is that the subject is just looking at a flickering light. We have now included videos of the flickering light as supplementary information (we show three different videos that are running on parallel: the signal for the digit “7” transmitted and reconstructed in Fig2, the “0”-“1” digit classification and also the signals for the PNN data shown in Figure 4). There is no way that someone can look at this and infer what the image is or what the classification data is for the machine learning task. Finally, the output data also is not an image but just a flickering response from neurons collected with one single EEG electrode from the visual cortex. There is no image information in this signal. Rather, we have encoded the information in the amplitudes of the different flicker frequencies, but the brain has no way of directly interpreting these signals. We hope this is also clear when looking at the video.

5. A technical question: what is the highest modulation frequency the human visual system can respond to? Is there a frequency transfer function?

REPLY: yes, there is indeed a frequency transfer function. This is basically an $\exp(-f)$ function and can be seen quite clearly for example in figure 2f (the exponential decay of the signal is visible also in the other figures but is clearest in f) and we now explicitly mention this in the manuscript. We are looking at the full visual system and we find that the SSVEP frequency spectrum, across every individual tested to date (also in other studies beyond this work) show an exponential transfer function of the kind $\exp(-f)$. This also matches well with the well-

known fact that SSVEP signals are weaker at higher frequencies and essentially zero above 50 Hz.

6. A technical question: The image encoding scheme seems to hide the correlation between adjacent pixels in a 2D image, and therefore worth more consideration. A natural question is: what if we permute all the pixels and encode them with different frequencies? The resultant light flickering pattern would be different, but it still represents the same image. I am curious to know how the classification results would be.

REPLY: we have indeed tried this. Because all frequencies are transmitted simultaneously, the specific rule with which frequencies are assigned to pixels and then transmitted does not make any difference to the final result. We comment on this in the revised paper.

7. A technical question: in some binary classification problems, such as differentiating 0 and 1, despite the flickering pattern, I think the total intensity of the light is definitely quite different if both images are normalized only to the brightest pixel in each image. This is because images of 0s tend to have many more bright pixels than images of 1s. I suspect humans can classify these two digits just based on average intensity, and therefore I would be curious to know what the result would be if images are normalized to have the same average intensity.

REPLY: we understand what the referee is saying. First of all, we note that the total intensity is actually renormalised before transmission so that is 8 cd for all experiments. This is quoted in the main text but is also now explicitly pointed out in the new “Methods” sections that we have added to describe the setup.

It is also important to note that the classification is not performed by the human participants. Human participants, could in principle, have noticed differences in the average luminance of a flicker stimulus. However, they did not have any knowledge about the digit encoded in the flicker stimulus. The classification is only performed by the computer that analyses the SSVEP frequency amplitudes, measured by EEG. These amplitudes are then always normalised before being processed. We do this normalisation mainly so that we can use the synthetic training data (which is normalised to one) and also to remove precisely any possible variability from person to person or within the same person maybe looking at different light amplitudes. So this normalisation effectively verifies the point raised by the referee, i.e. the results do not and cannot depend on average light intensities because any such possible variations are normalised out (before classification). We underline this point in the revised text.

8. Another technical question: I still find it hard to reconcile the fact that the level of attention affects classification performance while the nonlinear wave mixing model does not seem to vary as much. I wish to verify if attention changes the nonlinear wave mixing model (i.e., χ_2 , also it is also curious to understand why a single χ_2 works for all pairs of frequencies.).

REPLY: we believe that there may be a misunderstanding here. We comment in the paper that the opposite is true actually – we see that the nonlinear wave mixing *does* change with attention and that is used to explain the change also in classification. The two are most definitely connected. This possibly passed unobserved so we have underlined this more

strongly in the revised manuscript. We also know that the chi modulation function does not change appreciably from one subject to another and that this exponential decay is a universal function. This is true across our own data but it also matches with all results that can be found in literature, including the fact that SSVEP is not detectable above roughly 50Hz (see also reply above).

9. There are several typos in the manuscript. For example, there are two (m) panels in Figure 2, while one of them should be (g). There is 'network' spelled as 'netwoek', and so on. The authors might want to carefully check the manuscript and reference list again.

REPLY: thank you, we have fixed the figure and typos

Overall, the work is novel, and it would be a good contribution to the physical computing community if we can draw more definite conclusions on what we learned from these experiments.

Reviewer #3 (Remarks to the Author):

This paper proposed a method to encode and decode image data through BCI. While this paper presents interesting work and idea, the practicality of such method remains questionable.

The stimulation/recording time is too long (16 to 196 s) compared to traditional SSVEP-based BCI (typically <3-5 s). It will be even longer if the image is coloured and/or more complex (more pixels, higher resolution). The proposed method could be very useful in image transmission with no prior knowledge on the image. However, the current structure of the proposed approach still relies on image identification, which requires knowledge to the potential image set. In this case, labelling each possibility with a frequency and perform a traditional SSVEP task might be a more practical solution.

REPLY: we agree that 16 seconds or more acquisition times are long. We note however, that we did not perform this work with the goal of proposing a fast BCI. Rather the scope was more blue-sky enquiry into how much information can be encoded and if tasks such as machine learning can be performed directly with SSVEP. We find that question rather intriguing and the results rather surprising. One option moving forward could be to look at less trivial information encoding approaches, such as e.g. tagging a single frequency to a more complex piece of information. We now comment on this in the conclusions where we explain explicitly that the chosen applications are inspired by work in other fields such as physical neuromorphic computing.

This proposed method also poses a high requirement on the stimulation system, and the authors have not discussed this aspect in detail.

REPLY: We agree. We have added a sentence on this in the conclusions.

Comments:

1. The paper stressed on the low ITR in current solutions. But the proposed method did not solve this problem. The ITR from this work was not reported.

REPLY: we agree with the referee. We have revised the paper and explicitly point out that we can obtain images with a more than 12 bits/second rate, i.e. 12 times faster than standard SSVEP BCIs and more than 3 times faster than the previous record.

2. The authors should keep their notations consistent (e.g., on page 2 right below equation 1, the use of 'm' and 'M') and explain all parameters.

REPLY: corrected

3. When mentioning Fig.2f, the single period stimulation time should be longer than 16s. It is inaccurate and misleading to say $T=16s$.

REPLY: We are not sure that we understand this comment. T (the time required to transmit once all of the data is given by $1/\Delta f$, i.e. $1/(\text{bandwidth}/\text{number of frequencies})$). For figure 2f, this means $T=1/(12/196)=16$ seconds. So 16 seconds is the shortest possible acquisition time with these parameters and this is what we have reported.

4. The 196 s stimulation is really long. It is very hard for the users to stay focused on the visual stimulation for over 3 minutes. The stimulation may also cause severe visual fatigue.

REPLY: 196 seconds is relatively long but we use a red light at low intensity and there is no real fatigue over this length of time for a single measurement. We have noticed that if presented many times consecutively, then yes, fatigue accumulates. But this was used only in one instance just to showcase the effect of illumination time and certainly not as the final application scenario. We now underline this point in the text. We do indeed show the same results for 16 seconds, which we found to be very manageable. We should also bear in mind that BCI operators do often interact with SSVEP for long periods of time (if using this approach eg for typing text). And very typical SSVEP measurements are carried out over durations of several minutes. So in this context, our measurements are actually quite short. But we agree with the overall though and the text now comments on this.

5. Page 3 last sentence in the first paragraph: remove the extra ')'.

REPLY: corrected

6. Missing details on EEG system used in the experiment.

REPLY: the EEG is described in detail in a previous publication that we cite. We have added a description in the Methods.

7. Missing details on experimental setup, especially the stimulation system.

REPLY: this is provided in the Methods section

REVIEWER COMMENTS

Reviewer #1 (Remarks to the Author):

Although the authors have answered most of the questions, there are still some issues that require further clarification from the authors:

Gordon et al. mentioned that intermodulation frequencies appeared as the sum of any non-zero integer multiple of the input frequencies (e.g. $2f_1+f_2$, f_2-f_1 , etc.) The intermodulation frequency in this manuscript is only $f_1 \pm f_2$, and there are the same for all subjects, please discuss.

Gordon N, Hohwy J, Davidson MJ, van Boxtel JJA, Tsuchiya N. From intermodulation components to visual perception and cognition—a review. *Neuroimage*. 2019 Oct 1;199:480-494.

ITR is the most widely used metric in BCI performance evaluation. According to the calculation formula of ITR, ITR is affected by the number of choices, the accuracy of target detection, and the average time for a selection. I'm not sure how 12 bits/second is calculated in this manuscript.

Gao S, Wang Y, Gao X, Hong B. Visual and auditory brain-computer interfaces. *IEEE Trans Biomed Eng*. 2014 May;61(5):1436-47.

Reviewer #2 (Remarks to the Author):

I appreciate the authors for detailed answers to my questions.

After supplementing more control experiments, the experiments results are more rigorous and convincing. I support the publication of the results.

Reviewer #3 (Remarks to the Author):

I am not satisfied with the authors' response to my previous comments. The authors have marked that they corrected the manuscript in their response to some of the comments, however, no correction was found in the revised manuscript. The request to provide additional details in methods was also not taken seriously by the authors. Even though I stressed the importance of reporting the stimulation system in detail, including the system requirements and how they were set up, the authors failed in providing the details in their revised manuscript. It is also surprising to see the authors insist on $196/12=16$.

I do not think this manuscript meets the standard for publication in *Nature Communications* as the manuscript is not rigorous enough, did not present sufficient details in methods, and has clear limitations in its contribution.

Reviewer #3 (Remarks on code availability):

N/A

REPLY TO REVIEWER COMMENTS

Reviewer #1 (Remarks to the Author):

Although the authors have answered most of the questions, there are still some issues that require further clarification from the authors:

Gordon et al. mentioned that intermodulation frequencies appeared as the sum of any non-zero integer multiple of the input frequencies (e.g. $2f_1+f_2$, f_2-f_1 , etc.) The intermodulation frequency in this manuscript is only $f_1 \pm f_2$, and there are the same for all subjects, please discuss.

Gordon N, Hohwy J, Davidson MJ, van Boxtel JJA, Tsuchiya N. From intermodulation components to visual perception and cognition-a review. *Neuroimage*. 2019 Oct 1;199:480-494.

REPLY: We agree with the reviewer that, in general, all possible combinations of intermodulation frequencies might be observed in these experiments, and indeed, these are also observed in our experiments too. So, for example, if one looks at figure 2a, difference frequency terms f_2-f_1 are visible at very low frequencies (close to 0 Hz) and the “third order” terms e.g. $2f_1+f_2$ and similar combinations, can indeed also be seen at high frequencies i.e. centred around the third harmonic region $3*f_1$, of order 40 Hz. However, in this study, we focused only on the IM region centred around the second harmonic region and this is why we only discuss these terms in the description.

We have added the citation indicated by the reviewer and mentioned that, in general, difference frequency and high-order mixing terms in the third harmonic region can also be seen and might be considered in further studies.

ITR is the most widely used metric in BCI performance evaluation. According to the calculation formula of ITR, ITR is affected by the number of choices, the accuracy of target detection, and the average time for a selection. I'm not sure how 12 bits/second is calculated in this manuscript.

REPLY: We apologise for not including an explicit description of how the bit rate was calculated. We agree with the referee in terms of the elements that need to be considered. Specifically: “number of choices” in our case is black or white (0 or 1) values for each of the 196 pixels; “accuracy” corresponds to the pixel reconstruction error; and “time for selection” is the total measurement time for each image.

For a very first back-of-the-envelope estimate, ITR can be estimated by taking the final reconstructed image, which has $14 \times 14 = 196$ pixels and a 0 or 1 value for each pixel, i.e. 1 bit/pixel, equating to 196 bits information per image. For the case of figure 2(m), this was transmitted in 16.3 seconds, i.e. the information transmission rate for this image is $(\text{number of pixels}) \times (\text{bits/pixel}) / \text{time} = 12 \text{ bits/second}$.

This turns out to be relatively correct as the only missing element is the “accuracy” and indeed, the images are relatively well reconstructed. However, we can of course carry out a rigorous estimate by using the mutual information between the input and reconstructed images, which reflects the amount of information that was actually transmitted. This properly accounts also for accuracy. We provide full details of this in the SM and show two graphs that cover all of the experiments carried out for the imaging section. The maximum bit rate is found to be 10 bits/second.

We have now added a description of the approximate calculation in the main text alongside a mention that 12 bits/second was indeed an overestimate of the actually transmitted information and a reference to the actual maximum value of 10 bits/second with a full description and new figures in the SM.

Reviewer #2 (Remarks to the Author):

I appreciate the authors for detailed answers to my questions. After supplementing more control experiments, the experiments results are more rigorous and convincing. I support the publication of the results.

Reviewer #3 (Remarks to the Author):

I am not satisfied with the authors' response to my previous comments. The authors have marked that they corrected the manuscript in their response to some of the comments, however, no correction was found in the revised manuscript. The request to provide additional details in methods was also not taken seriously by the authors. Even though I stressed the importance of reporting the stimulation system in detail, including the system requirements and how they were set up, the authors failed in providing the details in their revised manuscript. It is also surprising to see the authors insist on $196/12=16$.

REPLY: We apologise for not understanding the reviewer's initial point about the 16 second measurement time. The origin of this is that for these measurements, the difference perceived by a human between a 16 second and a 16.3 second measurement time is negligible, so we simply rounded the number to the closest meaningful integer. In light of the reviewer's desire that we show all digits, we have corrected all instances in the manuscript to now read 16.3 seconds.

We are however a bit confused by the comment that no changes were made – these were all indicated in red text and were quite extensive (and visible to the other reviewers). We also included a section in the Methods dedicated to the light stimulation and a video showing how the stimulation was set up. In all honesty, this is an extremely simple setup and we believe that we have provided the details required to reproduce these tests.

I do not think this manuscript meets the standard for publication in Nature Communications as the manuscript is not rigorous enough, did not present sufficient details in methods, and has clear limitations in its contribution.

REPLY: The reviewer has not listed any specific missing information, but we have now added further extensive information in the Methods section, including e.g. a description of the exact relative location of the participant, details on the illumination, details on the measurement protocol that we followed, and a reference to a previous publication in which our EEG system was described in detail. This is all included in a new section in the "Methods". We now also provide a DOI with all of the data and also the codes used in this work.

As for the novelty, we respectfully disagree with this assessment – this work presents the fastest bit-rate transmission ever reported using EEG and then uses this system to report the first-ever demonstration that one can encode information across hundreds of individual frequency channels simultaneously and finally, using this to demonstrate neuromorphic classification capability. This latter idea has never, to the best of our knowledge, been proposed or demonstrated before. We then also demonstrate a route to enhance the neuromorphic processing capability by connecting more than one brain (participant) in series and this results in a significant improvement in the classification of small data sets. We sincerely believe that these results will be of great interest to a very broad audience and we have already seen this interest explicitly expressed by everyone who has seen this work.

REVIEWERS' COMMENTS

Reviewer #1 (Remarks to the Author):

The authors emphasize that this study achieves the fastest bit rate transmission to date. I am not satisfied with the author's response to my previous question on ITR calculation. ITR is an indicator commonly used to evaluate the BCI performance, but this manuscript does not use the conventional ITR calculation formula to calculate it, which may confuse readers and is not convenient for comparison with existing work performance.